# Capivasertib plus fulvestrant in patients with HR-positive/HER2-negative advanced breast cancer: phase 3 CAPItello-291 study extended Chinese cohort

In the global CAPItello-291 randomized phase 3 study (NCT04305496) in patients with hormone receptor-positive/HER2-negative advanced breast cancer and progression during/after aromatase inhibitor treatment, capivasertib–fulvestrant significantly improved progression-free survival (PFS) in the overall population and patients with *PIK3CA/AKT1/PTEN*-altered tumors versus placebo–fulvestrant. We assessed efficacy and safety of capivasertib–fulvestrant in a prespecified exploratory analysis of a Chinese cohort (*n* = 24) and extended study with the same protocol (*n* = 110). Clinically meaningful PFS benefit for capivasertib–fulvestrant was observed in the overall population (median PFS: 6.9 [capivasertib–fulvestrant] versus 2.8 [placebo–fulvestrant] months; hazard ratio 0.51, 95% CI 0.34–0.76), patients with *PIK3CA/AKT1/PTEN*-altered tumors (*n* = 46; 5.7 versus 1.9 months; hazard ratio 0.41, 95% CI 0.19–0.85) and *PIK3CA/AKT1/PTEN*-non-altered tumors (patients with confirmed next-generation sequencing results [*n* = 68]; 9.2 versus 2.7 months; hazard ratio 0.38; 95% CI 0.21–0.68). The most frequent adverse events (AEs) with capivasertib–fulvestrant were diarrhea (60.6% versus 11.3% with placebo–fulvestrant) and hyperglycemia (57.7% versus 17.7%). AEs leading to capivasertib–fulvestrant discontinuation were reported in 11.3% of patients versus 3.2% for placebo–fulvestrant. The benefit-risk profile of capivasertib–fulvestrant in the Chinese cohort was favorable; further exploration in patients with *PIK3CA/AKT1/PTEN*-non-altered tumors is warranted.

The incidence and mortality of breast cancer in China are rising. In 2020, breast cancer was the leading cancer diagnosed in women in China. Almost 420,000 cases and over 117,000 deaths were recorded in Chinese women and, as such, breast cancer in China accounted for 18.4% of the global cases and 17.1% of the global deaths[1]. Hormone receptor-positive/human epidermal growth factor receptor 2-negative (HR-positive/HER2-negative) breast cancer is the most commonly diagnosed subtype[2–5]. In 2019, approximately 54% of breast cancer diagnoses in China were HR-positive-/HER2-negative[5].

In certain patients with HR-positive/HER2-negative breast cancer, clinical guidelines recommend endocrine therapy (aromatase inhibitor) in combination with a cyclin-dependent kinase 4/6 (CDK4/6) inhibitor as first-line treatment for advanced disease[6–8]. Although the addition of CDK4/6 inhibitors to endocrine therapy results in significant clinical benefit, most patients will eventually experience

✉e-mail: xchu2009@hotmail.com

disease progression[9,10]. There is no uniform recommended standard-of-care treatment in patients who progress on endocrine therapy plus CDK4/6 inhibitor[9]; in recent randomized controlled studies, fulvestrant monotherapy in this population resulted in a median progression-free survival (PFS) of only 1.9–3.6 months[11–13].

The phosphatidylinositol-3-kinase (PI3K)/Akt serine/threonine kinase (AKT) pathway—a key signaling pathway controlling cell survival and metabolism—is frequently deregulated in cancer and has been the target of drug discovery[14]. Activating mutations in the catalytic subunit alpha of phosphatidylinositol-3-kinase (PIK3CA) and in AKT1, and inactivating alterations in the phosphatase and tensin homolog (PTEN), underpin overactivation of the pathway through genetic mechanisms in about half of HR-positive/HER2-negative breast cancers including in Chinese patients[15–24]. In addition, overactivation of the pathway has been noted in the tumors of patients who have experienced disease progression on endocrine therapy in the presence or absence of genetic tumor alterations[25–27].

Capivasertib is an oral inhibitor of AKT, a central node of the PI3K/AKT pathway. In vitro, capivasertib inhibits all AKT isoforms (AKT1/2/3) with a potency of ≤10 nmol/L, and AKT substrate phosphorylation with a potency of approximately 0.3–0.8 μmol/L[28]. A 4 days on, 3 days off dosing regimen of capivasertib was selected in the early clinical development process, to maximize the therapeutic window of AKT inhibition[29]. In the multicenter phase 2 FAKTION study, treatment with capivasertib–fulvestrant significantly improved PFS and overall survival, compared with placebo–fulvestrant, among postmenopausal patients with HR-positive/HER2-negative advanced breast cancer who had previously received endocrine therapy[30,31]. The global phase 3 CAPItello-291 study subsequently enrolled patients with HR-positive/HER2-negative advanced breast cancer and disease progression during or after treatment with aromatase inhibitors across 19 countries[13]. Compared with placebo–fulvestrant, the addition of capivasertib to fulvestrant resulted in statistically significant and clinically meaningful improvement in the dual primary endpoints of PFS in the overall population (hazard ratio 0.60, 95% confidence interval [CI] 0.51–0.71, P < 0.001) and in the population of patients with PIK3CA/AKT1/PTEN-altered tumors (hazard ratio 0.50, 95% CI 0.38–0.65, P < 0.001)[13]. These data led to the inclusion of capivasertib–fulvestrant as a treatment option in the United States National Comprehensive Cancer Network guidelines.

Here, we report the efficacy and safety of capivasertib–fulvestrant versus placebo–fulvestrant in a Chinese cohort of patients with HR-positive/HER2-negative advanced breast cancer whose disease progressed during or after aromatase inhibitor therapy and with or without a CDK4/6 inhibitor. The patients were recruited as part of either the global CAPItello-291 phase 3 study or, predominantly, in an extended Chinese cohort after recruitment to the global study had closed.

## Results

### Patients

In total, 134 patients (24 patients from the global population and 110 patients from an extended cohort) were recruited from 25 sites in mainland China (n = 118) and 3 National Medical Products Administration (NMPA)-certified sites in Taiwan (n = 16). Patients were randomized to capivasertib–fulvestrant (n = 71) or placebo–fulvestrant (n = 63; Supplementary Fig. 1). Results are reported for the overall cohort of 134 patients, but exploratory PFS analysis was also conducted in the extended cohort of 110 patients.

PIK3CA/AKT1/PTEN tumor alterations were detected in 24 patients (33.8%) receiving capivasertib–fulvestrant and in 22 patients (34.9%) receiving placebo–fulvestrant (Supplementary Table 1). PIK3CA/AKT1/PTEN tumor alterations were not detected in 35 patients (49.3%) receiving capivasertib–fulvestrant and 33 patients (52.4%) receiving placebo–fulvestrant, while tumor alteration status was unknown for 12 patients (16.9%) receiving capivasertib–fulvestrant and 8 patients (12.7%) receiving placebo–fulvestrant (Supplementary Table 1). Baseline characteristics were similar between treatment groups in the overall population and in patients with PIK3CA/AKT1/PTEN-altered tumors (Table 1). Overall, 27 patients (38.0%) receiving capivasertib–fulvestrant and 23 patients (36.5%) receiving placebo–fulvestrant had been previously treated with CDK4/6 inhibitor for advanced breast cancer (Table 1). The most common reason for no previous CDK4/6 inhibitor treatment was lack of affordability or reimbursement (68.3%).

### Treatment

At the primary analysis (data cutoff 8 May 2023), 18 patients (25.4%) were continuing to receive capivasertib and 10 patients (16.1%) were continuing to receive placebo. The main reason for discontinuation of capivasertib/placebo was radiological disease progression (Supplementary Fig. 1). In the capivasertib–fulvestrant group, patients were treated with capivasertib for a median of 5.5 months and with fulvestrant for a median of 6.0 months. In the placebo–fulvestrant group, patients received placebo for a median of 2.2 months and fulvestrant for a median of 2.8 months.

### Efficacy

The primary analysis for this cohort was conducted after 102 events of disease progression or death had occurred in the total study population (48 events in the capivasertib–fulvestrant group and 54 in the placebo–fulvestrant group) and 39 events of disease progression or death had occurred in the population of patients with PIK3CA/AKT1/PTEN-altered tumors (21 events in the capivasertib–fulvestrant group and 18 in the placebo–fulvestrant group). In the overall population of the Chinese cohort, the median PFS, as assessed by the investigator, was 6.9 months in the capivasertib–fulvestrant group versus 2.8 months in the placebo–fulvestrant group (hazard ratio 0.51, 95% CI 0.34–0.76; Fig. 1a). In the population of patients with PIK3CA/AKT1/PTEN-altered tumors, the median PFS, as assessed by the investigator, was 5.7 months in the capivasertib–fulvestrant group versus 1.9 months in the placebo–fulvestrant group (hazard ratio 0.41, 95% CI 0.19–0.85; Fig. 1b). In patients with PIK3CA/AKT1/PTEN-non-altered tumors, a PFS benefit was observed in the capivasertib–fulvestrant group versus the placebo–fulvestrant group (hazard ratio 0.56, 95% CI 0.34–0.94, including patients with unknown next-generation sequencing [NGS] results [Supplementary Fig. 2a]; hazard ratio 0.38, 95% CI 0.21–0.68, excluding patients with unknown NGS results [Supplementary Fig. 2b]). The hazard ratio was not calculated for patients with an unknown NGS result due to an insufficient number of events (Supplementary Fig. 2c).

Consistent benefit with capivasertib–fulvestrant compared with placebo–fulvestrant was observed in clinically relevant subgroups, including in patients with visceral metastases and in those who received prior CDK4/6 inhibitor treatment or prior chemotherapy for advanced breast cancer (Fig. 2). Benefit was also seen in clinically relevant subgroups of patients with PIK3CA/AKT1/PTEN-altered tumors, but the 95% CIs were wider and the hazard ratios not calculated in some instances, because of the small sample size (Supplementary Fig. 3). PFS benefit (as assessed by blinded independent central review) in the overall population and in the population of patients with PIK3CA/AKT1/PTEN-altered tumors was broadly consistent with that assessed by the investigator (Supplementary Fig. 4).

In a post hoc exploratory analysis in the extended cohort only (n = 110), the median PFS, as assessed by the investigator, was 6.3 months in the capivasertib–fulvestrant group versus 3.0 months in the placebo–fulvestrant group (hazard ratio 0.51, 95% CI 0.33–0.80; Supplementary Fig. 5a). In the population of patients with PIK3CA/AKT1/PTEN-altered tumors (n = 37), the median PFS was 5.7 months in the capivasertib–fulvestrant group versus 1.9 months in the

**Table 1 | Characteristics of the Chinese cohort at baseline**

| Characteristic | | Overall population | | Patients with *PIK3CA/AKT1/PTEN*-altered tumors | |
|---|---|---|---|---|---|
| | | Capivasertib–fulvestrant (*n* = 71) | Placebo–fulvestrant (*n* = 63) | Capivasertib–fulvestrant (*n* = 24) | Placebo–fulvestrant (*n* = 22) |
| Age, median (range), years | | 56.0 (26–82) | 54.0 (34–72) | 55.5 (32–73) | 55.0 (34–72) |
| Female sex, *n* (%) | | 71 (100) | 62 (98.4) | 24 (100) | 22 (100) |
| Postmenopausal, *n* (%) | | 50 (70.4) | 43 (68.3) | 19 (79.2) | 15 (68.2) |
| ECOG/WHO PS, *n* (%) | 0 | 35 (49.3) | 31 (49.2) | 11 (45.8) | 13 (59.1) |
| | 1 | 36 (50.7) | 32 (50.8) | 13 (54.2) | 9 (40.9) |
| Race, *n* (%) | Asian | 69[a] (100) | 61[a] (100) | 24 (100) | 22 (100) |
| Sites of metastases, *n* (%) | Bone only | 5 (7.0) | 4 (6.3) | 1 (4.2) | 1 (4.5) |
| | Visceral | 52 (73.2) | 49 (77.8) | 21 (87.5) | 18 (81.8) |
| | Liver | 32 (45.1) | 29 (46.0) | 14 (58.3) | 10 (45.5) |
| Stage at diagnosis, *n* (%) | Stage 0–IIIC | 52 (73.2) | 47 (74.6) | 19 (79.2) | 19 (86.4) |
| | Stage IV | 15 (21.1) | 10 (15.9) | 5 (20.8) | 2 (9.1) |
| Hormone receptor status, *n* (%) | ER-positive/PgR-positive | 51 (71.8) | 42 (66.7) | 15 (62.5) | 14 (63.6) |
| | ER-positive/PgR-negative | 20 (28.2) | 21 (33.3) | 9 (37.5) | 8 (36.4) |
| Endocrine status, *n* (%) | Primary resistance | 27 (38.0) | 26 (41.3) | 7 (29.2) | 12 (54.5) |
| | Secondary resistance | 44 (62.0) | 37 (58.7) | 17 (70.8) | 10 (45.5) |
| Prior therapies for advanced breast cancer, *n* (%) | 0 | 15 (21.1) | 10 (15.9) | 3 (12.5) | 6 (27.3) |
| | 1 | 31 (43.7) | 35 (55.6) | 13 (54.2) | 10 (45.5) |
| | 2 | 22 (31.0) | 16 (25.4) | 7 (29.2) | 6 (27.3) |
| | 3 | 3 (4.2) | 2 (3.2) | 1 (4.2) | 0 |
| Previous CDK4/6 inhibitor, *n* (%) | (Neo)adjuvant only | 1 (1.4) | 1 (1.6) | 0 | 1 (4.5) |
| | Advanced breast cancer | 27 (38.0) | 23 (36.5) | 11 (45.8) | 7 (31.8) |
| Previous chemotherapy, *n* (%) | (Neo)adjuvant only | 36 (50.7) | 36 (57.1) | 14 (58.3) | 15 (68.2) |
| | Advanced breast cancer | 24 (33.8) | 16 (25.4) | 8 (33.3) | 5 (22.7) |

*AKT1* Akt serine/threonine kinase 1, *CDK4/6* cyclin-dependent kinase 4/6, *ECOG/WHO PS* Eastern Cooperative Oncology Group/World Health Organization performance status, *ER* estrogen receptor, *PgR* progesterone receptor, *PIK3CA* catalytic subunit alpha of phosphatidylinositol-3-kinase, *PTEN* phosphatase and tensin homolog.
[a]Data not available for two patients in each group.

placebo–fulvestrant group (hazard ratio 0.49, 95% CI 0.21–1.09; Supplementary Fig. 5b).

The objective response rate (as assessed by the investigator) was 29.4% in the capivasertib–fulvestrant group compared with 8.3% in the placebo–fulvestrant group (odds ratio 4.58, 95% CI 1.60–13.15; Supplementary Table 2). In patients with *PIK3CA/AKT1/PTEN*-altered tumors, the objective response rate (as assessed by the investigator) was 37.5% in the capivasertib–fulvestrant group compared with 9.1% in the placebo–fulvestrant group (the odds ratio was non-calculable, due to an insufficient number of responders; Supplementary Table 2). Best overall responses as assessed by the investigator and by blinded independent central review were similar (Supplementary Table 2).

At the time of analysis, 26 patients receiving capivasertib–fulvestrant (11 patients with *PIK3CA/AKT1/PTEN*-altered tumors) and 28 patients receiving placebo–fulvestrant (10 patients with *PIK3CA/AKT1/PTEN*-altered tumors) had progressed on the next subsequent therapy after discontinuation of study treatment or had died from any cause (Supplementary Fig. 6). Median progression-free survival 2 (PFS2) in the overall population was 15.0 months in the capivasertib–fulvestrant group versus 12.4 months in the placebo–fulvestrant group (hazard ratio 0.79, 95% CI 0.46–1.36; Supplementary Fig. 6a). Median PFS2 in the population of patients with *PIK3CA/AKT1/PTEN*-altered tumors was 14.6 months in the capivasertib–fulvestrant versus 8.8 months in the placebo–fulvestrant group (hazard ratio 0.91, 95% CI 0.38–2.21; Supplementary Fig. 6b).

Overall, 10 patients (14.1%) receiving capivasertib–fulvestrant and 16 patients (25.4%) receiving placebo–fulvestrant had died. Median overall survival was not reached in either the overall population or in the population of patients with *PIK3CA/AKT1/PTEN*-altered tumors (Fig. 3). The hazard ratio for the overall population was 0.48 (95% CI 0.21–1.05) and non-calculable for the population of patients with *PIK3CA/AKT1/PTEN*-altered tumors, due to an insufficient number of events.

**Subsequent therapies**
Overall, 40 patients (56.3%) in the capivasertib–fulvestrant group and 45 patients (71.4%) in the placebo–fulvestrant group received subsequent anticancer therapies following study treatment discontinuation (Supplementary Table 3). The most common therapies received were cytotoxic chemotherapy (40.8% in the capivasertib–fulvestrant group and 57.1% in the placebo–fulvestrant group), targeted therapy (22.5% in the capivasertib–fulvestrant group and 25.4% in the placebo–fulvestrant group), and hormonal therapy (21.1% in the capivasertib–fulvestrant group and 22.2% in the placebo–fulvestrant group). Subsequent therapies in the population of patients with *PIK3CA/AKT1/PTEN*-altered tumors were similar to those in the overall population (Supplementary Table 3).

**Safety**
The safety population included 71 patients who received capivasertib–fulvestrant and 62 patients who received placebo–

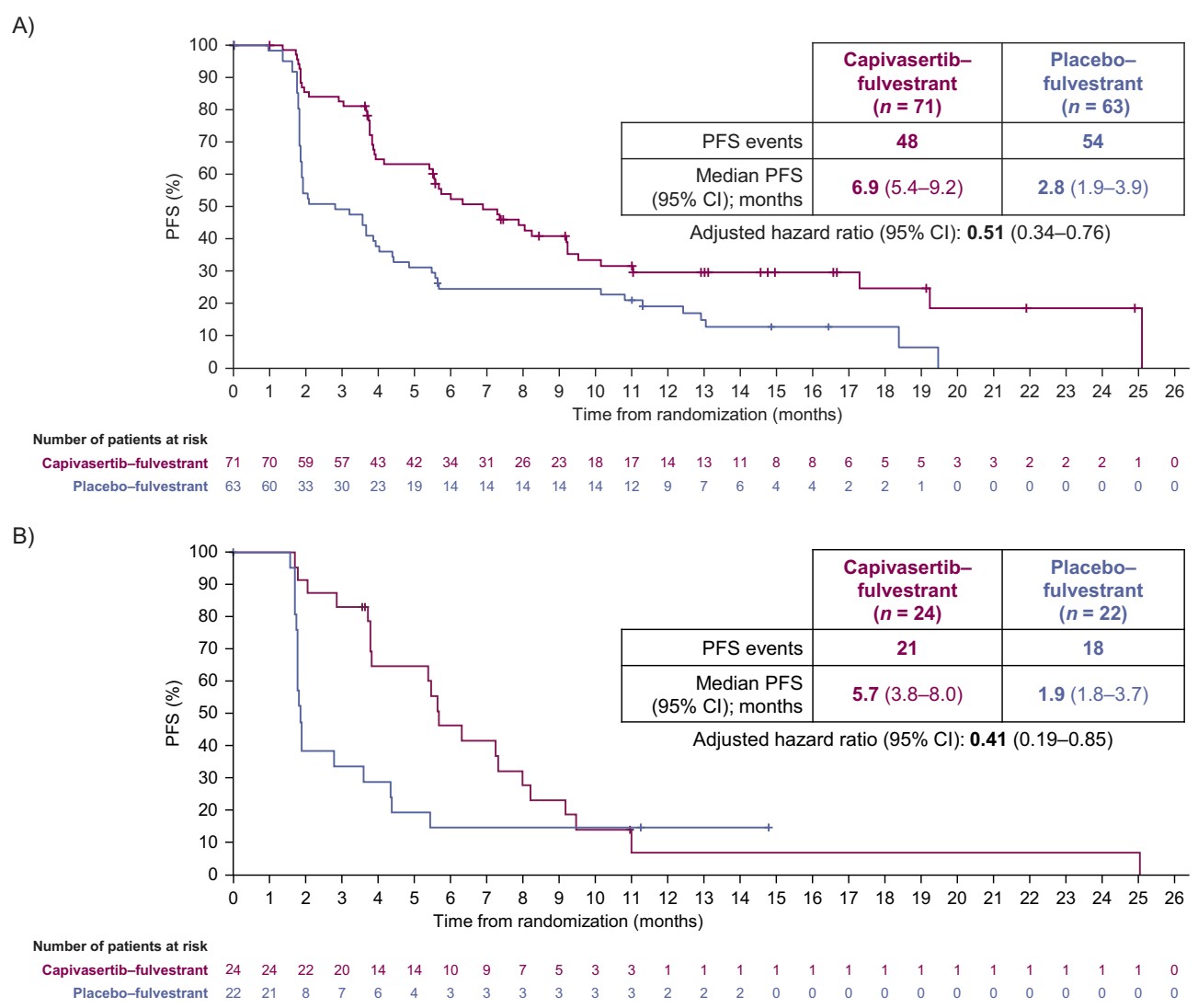

**Fig. 1 | Investigator-assessed PFS.** Reported for **A**, the Chinese cohort overall population and **B**, patients with *PIK3CA/AKT1/PTEN*-altered tumors. The hazard ratio of the capivasertib–fulvestrant (purple) and placebo–fulvestrant (blue) curves was estimated with the use of the Cox proportional hazards model, with stratification according to the presence or absence of liver metastases and previous use of a CDK4/6 inhibitor (yes or no) in the overall population, and according to previous fulvestrant. The most commonly reported adverse events (AEs) of any CDK4/6 inhibitor use in the population of patients with *PIK3CA/AKT1/PTEN*-altered tumors. Tick marks indicate censored data. *AKT1* Akt serine/threonine kinase 1, CDK4/6 cyclin-dependent kinase 4/6, CI confidence interval, *PIK3CA* catalytic subunit alpha of phosphatidylinositol-3-kinase, PFS progression-free survival, *PTEN* phosphatase and tensin homolog.

fulvestrant. The most commonly reported adverse events (AEs) of any grade in the capivasertib-fulvestrant group were diarrhea (60.6% versus 11.3% in the placebo-fulvestrant group) and hyperglycemia (57.7% versus 17.7%; Table 2). Rash (as a group term) occurred in 36 patients (50.7%) in the capivasertib-fulvestrant group and in 6 patients (9.7%) in the placebo-fulvestrant group. The most frequently reported AEs of grade ≥ 3 in the capivasertib-fulvestrant group were rash (15.5% versus 0% in the placebo-fulvestrant group), diarrhea (7.0% versus 0%) and hypokalemia (5.6% versus 4.8%). The safety profile of capivasertib-fulvestrant in the population of patients with *PIK3CA/AKT1/PTEN*-altered tumors was comparable with that in the overall population.

Serious AEs were reported in 20 patients (28.2%) receiving capivasertib-fulvestrant and in 3 patients (4.8%) receiving placebo-fulvestrant (Supplementary Table 4). Death due to AEs was reported in one patient (1.4%) in the capivasertib-fulvestrant group and in 0 patients in the placebo-fulvestrant group; the reported death was from renal failure and was not considered by the investigator to be related to capivasertib or fulvestrant. AEs led to discontinuation of capivasertib/placebo in eight patients (11.3%) receiving capivasertib

and in two patients (3.2%) receiving placebo (Supplementary Table 5), and led to dose reduction of capivasertib/placebo in 13 (18.3%) patients receiving capivasertib and in 0 patients receiving placebo (Supplementary Table 5). Dose interruptions were required for capivasertib/placebo in 36 patients (50.7%) receiving capivasertib and in 18 patients (29.0%) receiving placebo (Supplementary Table 5). Hyperglycemia AEs led to discontinuation of capivasertib/placebo in one patient (1.4%) receiving capivasertib and no patients receiving placebo, and to dose interruption in two patients (2.8%) receiving capivasertib and no patients receiving placebo. No dose reduction of capivasertib/placebo was required due to hyperglycemia AEs.

## Quality of life
European Organisation for Research and Treatment of Cancer Quality of Life Questionnaire Core 30 (EORTC QLQ-C30) scores were maintained from baseline and were similar between treatment arms throughout visits (Supplementary Fig. 7). Overall compliance rates were 94.4% in the capivasertib-fulvestrant group and 92.1% in the placebo-fulvestrant group. Compliance rates in the capivasertib-

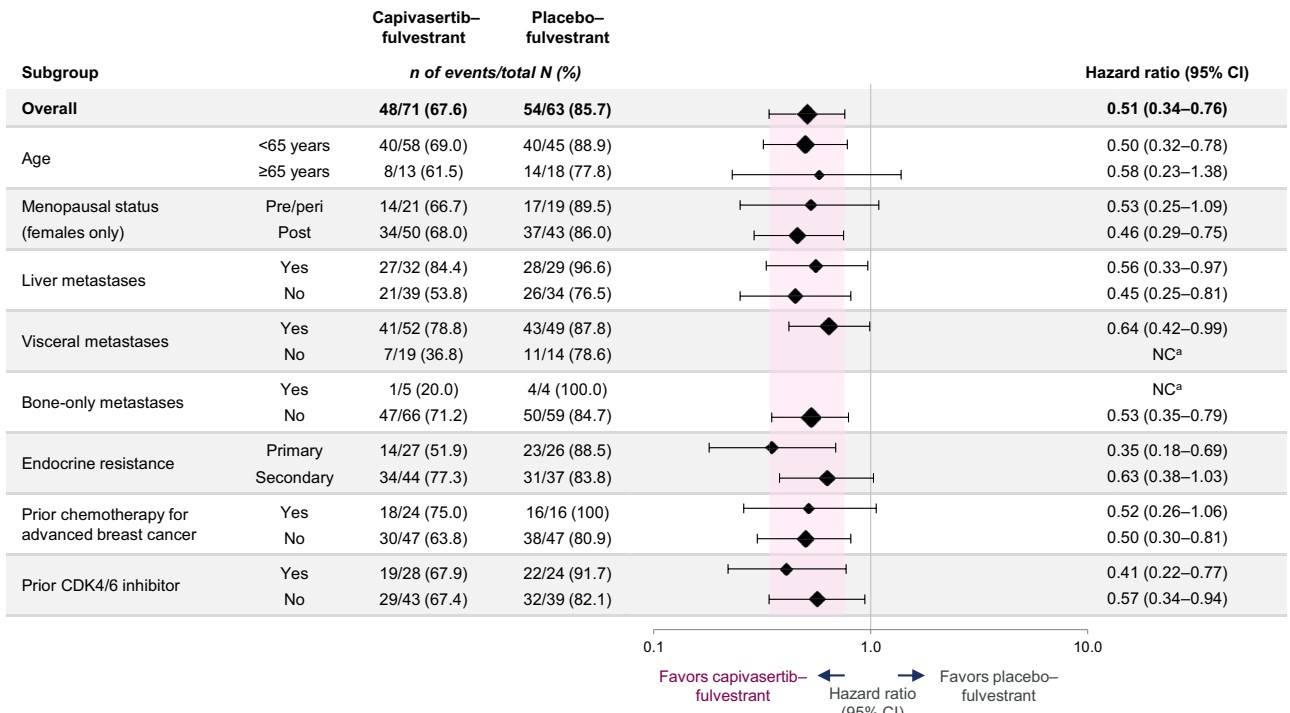

| Subgroup | | Capivasertib–fulvestrant | Placebo–fulvestrant | | Hazard ratio (95% CI) |
|---|---|---|---|---|---|
| | | n of events/total N (%) | | | |
| Overall | | 48/71 (67.6) | 54/63 (85.7) | | 0.51 (0.34–0.76) |
| Age | <65 years | 40/58 (69.0) | 40/45 (88.9) | | 0.50 (0.32–0.78) |
| | ≥65 years | 8/13 (61.5) | 14/18 (77.8) | | 0.58 (0.23–1.38) |
| Menopausal status (females only) | Pre/peri | 14/21 (66.7) | 17/19 (89.5) | | 0.53 (0.25–1.09) |
| | Post | 34/50 (68.0) | 37/43 (86.0) | | 0.46 (0.29–0.75) |
| Liver metastases | Yes | 27/32 (84.4) | 28/29 (96.6) | | 0.56 (0.33–0.97) |
| | No | 21/39 (53.8) | 26/34 (76.5) | | 0.45 (0.25–0.81) |
| Visceral metastases | Yes | 41/52 (78.8) | 43/49 (87.8) | | 0.64 (0.42–0.99) |
| | No | 7/19 (36.8) | 11/14 (78.6) | | NC[a] |
| Bone-only metastases | Yes | 1/5 (20.0) | 4/4 (100.0) | | NC[a] |
| | No | 47/66 (71.2) | 50/59 (84.7) | | 0.53 (0.35–0.79) |
| Endocrine resistance | Primary | 14/27 (51.9) | 23/26 (88.5) | | 0.35 (0.18–0.69) |
| | Secondary | 34/44 (77.3) | 31/37 (83.8) | | 0.63 (0.38–1.03) |
| Prior chemotherapy for advanced breast cancer | Yes | 18/24 (75.0) | 16/16 (100) | | 0.52 (0.26–1.06) |
| | No | 30/47 (63.8) | 38/47 (80.9) | | 0.50 (0.30–0.81) |
| Prior CDK4/6 inhibitor | Yes | 19/28 (67.9) | 22/24 (91.7) | | 0.41 (0.22–0.77) |
| | No | 29/43 (67.4) | 32/39 (82.1) | | 0.57 (0.34–0.94) |

Favors capivasertib–fulvestrant ← Hazard ratio (95% CI) → Favors placebo–fulvestrant

**Fig. 2 | Subgroup analysis of investigator-assessed PFS in the Chinese cohort overall population.** Subgroup analysis within the overall population was performed at each subgroup level with the use of a Cox proportional hazards model, including the trial-group term only. Data are presented as unstratified HRs and 95% CI. Selected subgroups of interest are shown. Menopausal status was assessed in women only. [a]Hazard ratio and 95% CI were not calculated due to an insufficient number of events (<20 across treatment groups). CDK4/6 cyclin-dependent kinase 4/6, CI confidence interval, NC not calculated, PFS progression-free survival.

fulvestrant group were ≥ 90% up to cycle 3, ≥ 80% at cycles 4 and 5, ≥ 75% at cycle 6 and ≥ 60% at cycles 7–10. The analysis was limited by the low compliance rate in the placebo–fulvestrant group, which decreased to < 60% by cycle 4 and was further reduced in subsequent cycles. The median time to deterioration (defined as a sustained decrease of ≥ 10 points in the score from baseline) in global health status/quality of life in the overall population was similar between treatment groups, amounting to 11.0 months in the capivasertib–fulvestrant group compared with 7.4 months in the placebo–fulvestrant group (hazard ratio 1.01, 95% CI 0.59–1.75).

## Discussion

In this phase 3, double-blind, randomized study in a Chinese cohort, a clinically meaningful improvement in PFS with capivasertib–fulvestrant compared with placebo–fulvestrant was seen in patients with HR-positive/HER2-negative advanced breast cancer whose disease had progressed during or after previous treatment with an aromatase inhibitor. The PFS benefit was seen in both the overall population and in the population of patients with PIK3CA/AKT1/PTEN-altered tumors (dual primary endpoint). This is the first randomized clinical dataset to demonstrate the clinical benefit of capivasertib–fulvestrant in a Chinese cohort of patients with HR-positive/HER2-negative advanced breast cancer; most patients were part of a dedicated extended study cohort for whom data have not been previously published.

These results are consistent with evidence from the phase 2 FAKTION[30,31] and global phase 3 CAPItello-291[13] studies. Additionally, consistent with the results in the global CAPItello-291 study[13], clinically meaningful benefit was observed across clinically relevant subgroups in the Chinese population, including patients with liver metastases and those previously treated with a CDK4/6 inhibitor or chemotherapy in the advanced setting, although the small sample size limited some analyses. A *post hoc* exploratory analysis in the extended Chinese cohort only (i.e. excluding patients recruited during the global

CAPItello-291 study) was consistent with the primary PFS analysis in the overall Chinese cohort, demonstrating a PFS benefit with capivasertib–fulvestrant over placebo–fulvestrant in both the overall extended Chinese cohort population and in the population of patients with PIK3CA/AKT1/PTEN-altered tumors. Although characteristics appeared broadly balanced between treatment groups, the subgroup findings from this study should be viewed with the caveat that no specific analyses were conducted to confirm this assumption.

The addition of capivasertib to fulvestrant also showed a PFS benefit in patients with PIK3CA/AKT1/PTEN-non-altered tumors. In the Chinese cohort, there was a more pronounced benefit in the group of patients with tumors with confirmed PIK3CA/AKT1/PTEN-non-altered status compared with the equivalent population in the global cohort of CAPItello-291[13]. However, comparisons are limited by population variability, the small sample size in the Chinese cohort, and the exploratory nature of these analyses. Nevertheless, PI3K/AKT pathway activation in cancer can also be enhanced by mechanisms other than alterations in PIK3CA/AKT1/PTEN[14,25,26], supported by preclinical data showing that capivasertib inhibited growth in some cell lines without PIK3CA/AKT1/PTEN alterations[28]. Notably, in an exploratory analysis of the global BOLERO-2 study, PFS benefit was observed with the combination of everolimus (a mechanistic target of rapamycin [mTOR] inhibitor) and exemestane versus placebo–exemestane in patients with HR-positive/HER-2 negative advanced breast cancer previously treated with non-steroidal aromatase inhibitors (without CDK4/6 inhibitors) and with a normal (not hyperactive) PI3K pathway activity[32]. Additionally, exploratory biomarker analyses in the neoadjuvant triple-negative breast cancer setting showed that high levels of phosphorylated AKT1 in the absence of PIK3CA/AKT/PTEN alterations were associated with other pathway activation mechanisms, including deficiency of PTEN protein expression and enhanced activation of upstream receptor tyrosine kinases[33]. Finally, crosstalk between the AKT and estrogen receptor pathways can reciprocally compensate monotherapy treatment in certain tumors, indicating that simultaneous

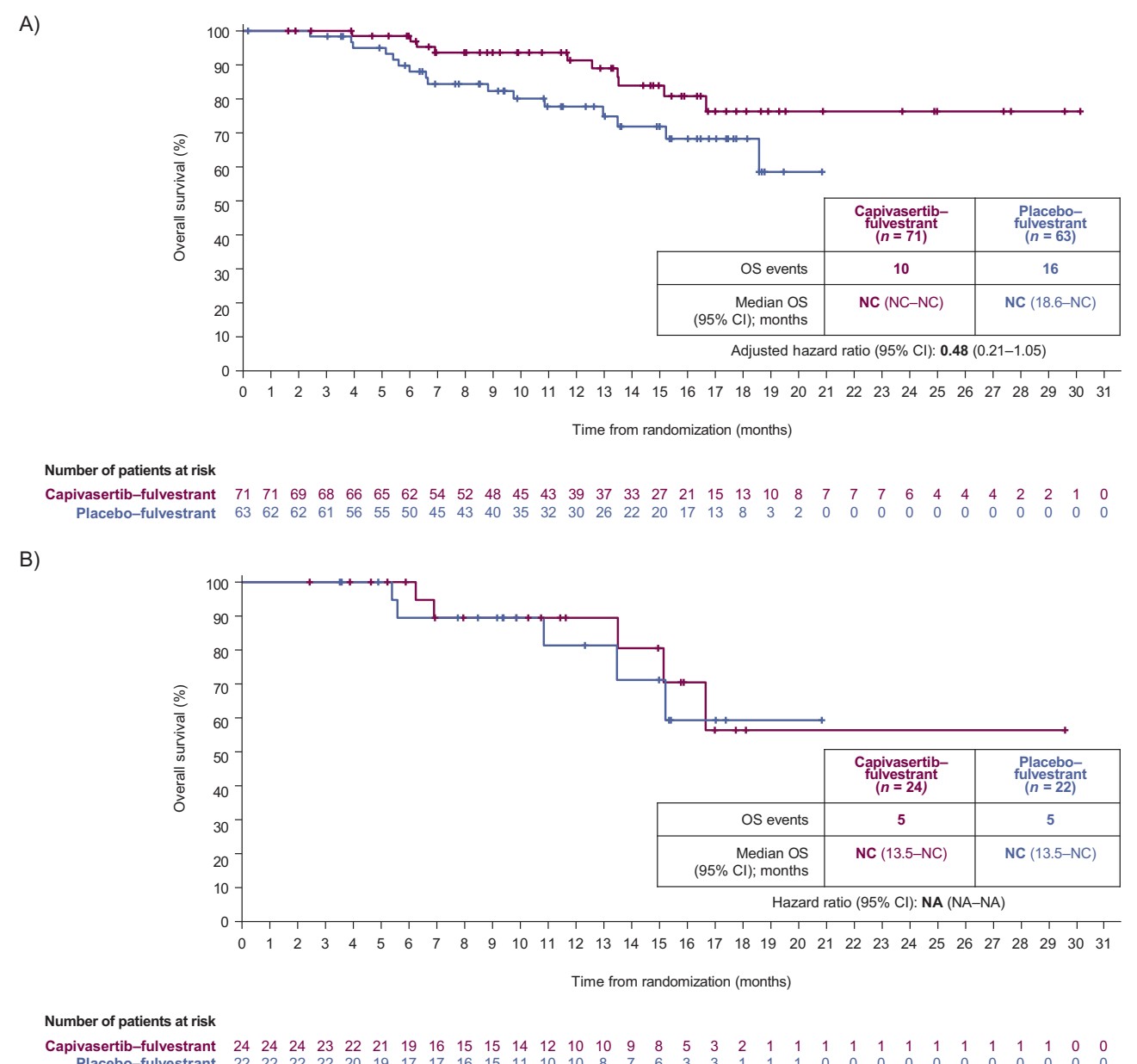

**Fig. 3 | Overall survival.** Reported for **A**, the Chinese cohort overall population and **B**, patients with *PIK3CA/AKT1/PTEN*-altered tumors. The hazard ratio of the capivasertib–fulvestrant (purple) and placebo–fulvestrant (blue) curves was estimated with the use of the Cox proportional hazards model, with stratification according to the previous use of a CDK4/6 inhibitor (yes or no) in the overall population. Results for the population of patients with *PIK3CA/AKT1/PTEN*-altered tumors are not presented due to an insufficient number of events (<20 across treatment groups). A sufficient number of deaths for a formal analysis of overall survival had not occurred by the data cutoff date (8 May 2023). Tick marks indicate censored data. A 0.01% alpha penalty was assigned to the overall survival analyses of no detriment (i.e. with the hazard ratio not favoring the placebo–fulvestrant group). *AKT1* Akt serine/threonine kinase 1, CDK4/6 cyclin-dependent kinase 4/6, CI confidence interval, NA non-applicable, NC not calculated, OS overall survival, *PIK3CA* catalytic subunit alpha of phosphatidylinositol-3-kinase, PFS progression-free survival, *PTEN* phosphatase and tensin homolog.

inhibition of both pathways could have an impact in the overall population independent of alteration status[34–36]. Hence, further research in Chinese patients with *PIK3CA/AKT1/PTEN*-non-altered tumors could be conducted to determine mechanisms, apart from alterations in *PIK3CA/AKT1/PTEN*, which could be associated with the PFS benefit exerted by capivasertib–fulvestrant.

The baseline characteristics of the patients in the Chinese cohort were generally representative of Chinese patients with HR-positive/HER2-negative advanced breast cancer[37,38]. Compared with the population in the global part of the study[13], more patients in the Chinese cohort had previously received chemotherapy for advanced breast cancer (29.9% versus 18.2%), and fewer had received CDK4/6 inhibitors

for advanced breast cancer (37.3% versus 69.1%). These findings are reflective of practice in China at the time of the study[5], but the treatment landscape has changed with the wider availability of CDK4/6 inhibitors (abemaciclib, dalpiciclib, palbociclib and ribociclib) in China[39].

The proportion of patients with *PIK3CA*-altered tumors was slightly lower in the Chinese cohort than that expected, based on recent literature[19–24] and compared with the global study[13]. This was potentially due to a competing clinical study concurrently recruiting patients with similar characteristics and *PIK3CA*-altered tumors at the same participating investigational centers. The study findings were not affected, as consistent benefit was observed in the overall population

**Table 2 | Most frequent AEs in the Chinese cohort overall population (safety population)[a]**

| AE, n (%) | Capivasertib–fulvestrant (n = 71) | | | | | | Placebo–fulvestrant (n = 62) | | | | | |
|---|---|---|---|---|---|---|---|---|---|---|---|---|
| | Any grade | Grade 1 | Grade 2 | Grade 3 | Grade 4 | Grade 5 | Any grade | Grade 1 | Grade 2 | Grade 3 | Grade 4 | Grade 5 |
| Any AE | 67 (94.4) | 12 (16.9) | 20 (28.2) | 31 (43.7) | 3 (4.2) | 1 (1.4)[b] | 56 (90.3) | 19 (30.6) | 29 (46.8) | 6 (9.7) | 2 (3.2) | 0 |
| Diarrhea | 43 (60.6) | 29 (40.8) | 9 (12.7) | 5 (7.0) | 0 | 0 | 7 (11.3) | 7 (11.3) | 0 | 0 | 0 | 0 |
| Hyperglycemia | 41 (57.7) | 30 (42.3) | 10 (14.1) | 0 | 1 (1.4) | 0 | 11 (17.7) | 11 (17.7) | 0 | 0 | 0 | 0 |
| Rash[c] | 36 (50.7) | 13 (18.3) | 12 (16.9) | 11 (15.5) | 0 | 0 | 6 (9.7) | 4 (6.5) | 2 (3.2) | 0 | 0 | 0 |
| Anemia | 22 (31.0) | 10 (14.1) | 10 (14.1) | 2 (2.8) | 0 | 0 | 12 (19.4) | 6 (9.7) | 5 (8.1) | 1 (1.6) | 0 | 0 |
| Hypokalemia | 18 (25.4) | 13 (18.3) | 1 (1.4) | 4 (5.6) | 0 | 0 | 6 (9.7) | 2 (3.2) | 1 (1.6) | 3 (4.8) | 0 | 0 |
| Weight decreased | 17 (23.9) | 12 (16.9) | 5 (7.0) | 0 | 0 | 0 | 3 (4.8) | 2 (3.2) | 0 | 1 (1.6) | 0 | 0 |
| Hypertriglyceridemia | 16 (22.5) | 9 (12.7) | 4 (5.6) | 3 (4.2) | 0 | 0 | 4 (6.5) | 4 (6.5) | 0 | 0 | 0 | 0 |
| Pyrexia | 16 (22.5) | 9 (12.7) | 7 (9.9) | 0 | 0 | 0 | 3 (4.8) | 3 (4.8) | 0 | 0 | 0 | 0 |
| Urinary tract infection | 15 (21.1) | 4 (5.6) | 10 (14.1) | 1 (1.4) | 0 | 0 | 12 (19.4) | 3 (4.8) | 9 (14.5) | 0 | 0 | 0 |
| White blood cell count decreased | 15 (21.1) | 4 (5.6) | 10 (14.1) | 1 (1.4) | 0 | 0 | 5 (8.1) | 3 (4.8) | 2 (3.2) | 0 | 0 | 0 |
| Neutrophil count decreased | 14 (19.7) | 6 (8.5) | 6 (8.5) | 2 (2.8) | 0 | 0 | 4 (6.5) | 3 (4.8) | 1 (1.6) | 0 | 0 | 0 |
| Alanine aminotransferase increased | 13 (18.3) | 12 (16.9) | 0 | 1 (1.4) | 0 | 0 | 21 (33.9) | 18 (29.0) | 3 (4.8) | 0 | 0 | 0 |
| Glycosylated hemoglobin increased | 13 (18.3) | 13 (18.3) | 0 | 0 | 0 | 0 | 3 (4.8) | 3 (4.8) | 0 | 0 | 0 | 0 |
| Proteinuria | 12 (16.9) | 11 (15.5) | 1 (1.4) | 0 | 0 | 0 | 5 (8.1) | 4 (6.5) | 1 (1.6) | 0 | 0 | 0 |
| Nausea | 11 (15.5) | 8 (11.3) | 2 (2.8) | 1 (1.4) | 0 | 0 | 2 (3.2) | 2 (3.2) | 0 | 0 | 0 | 0 |
| Aspartate aminotransferase increased | 10 (14.1) | 9 (12.7) | 1 (1.4) | 0 | 0 | 0 | 18 (29.0) | 14 (22.6) | 3 (4.8) | 1 (1.6) | 0 | 0 |
| COVID-19 | 10 (14.1) | 5 (7.0) | 5 (7.0) | 0 | 0 | 0 | 5 (8.1) | 5 (8.1) | 0 | 0 | 0 | 0 |
| Hypophosphatemia | 9 (12.7) | 8 (11.3) | 1 (1.4) | 0 | 0 | 0 | 3 (4.8) | 3 (4.8) | 0 | 0 | 0 | 0 |
| Stomatitis | 9 (12.7) | 5 (7.0) | 4 (5.6) | 0 | 0 | 0 | 0 | 0 | 0 | 0 | 0 | 0 |
| Blood creatinine increased | 8 (11.3) | 4 (5.6) | 4 (5.6) | 0 | 0 | 0 | 2 (3.2) | 2 (3.2) | 0 | 0 | 0 | 0 |
| Blood thyroid stimulating hormone increased | 8 (11.3) | 8 (11.3) | 0 | 0 | 0 | 0 | 5 (8.1) | 5 (8.1) | 0 | 0 | 0 | 0 |
| Decreased appetite | 8 (11.3) | 6 (8.5) | 2 (2.8) | 0 | 0 | 0 | 3 (4.8) | 1 (1.6) | 2 (3.2) | 0 | 0 | 0 |
| Hypercholesterolemia | 8 (11.3) | 7 (9.9) | 1 (1.4) | 0 | 0 | 0 | 4 (6.5) | 4 (6.5) | 0 | 0 | 0 | 0 |
| Hypoalbuminemia | 8 (11.3) | 7 (9.9) | 1 (1.4) | 0 | 0 | 0 | 7 (11.3) | 5 (8.1) | 2 (3.2) | 0 | 0 | 0 |
| Hypocalcemia | 8 (11.3) | 4 (5.6) | 4 (5.6) | 0 | 0 | 0 | 3 (4.8) | 3 (4.8) | 0 | 0 | 0 | 0 |
| Vomiting | 8 (11.3) | 6 (8.5) | 0 | 2 (2.8) | 0 | 0 | 3 (4.8) | 3 (4.8) | 0 | 0 | 0 | 0 |

AE adverse event, COVID-19 coronavirus disease 2019.
[a]The safety population included all the patients who received at least one dose of capivasertib, fulvestrant or placebo. The listed AEs were reported as a single term in at least 10% of the patients for any grade in either group. AEs are reported regardless of the relationship to the study drugs. [b]Death from renal failure not considered by the investigator to be related to capivasertib or fulvestrant. Renal failure is not included in the table as a preferred term, as the frequency of AEs of any grade was below the threshold of at least 10% in either group (n = 2 [2.8%] in the capivasertib–fulvestrant group; n = 0 in the placebo–fulvestrant group). [c]Group term of rash includes the preferred terms of rash, rash macular, rash maculopapular, rash papular and rash pruritic.

and in patients with *PIK3CA/AKT1/PTEN*-altered tumors in the Chinese cohort and the global population. Additionally, these findings were consistent despite the use of two different NGS detection platforms: (1) the OncoScreen Plus assay (Burning Rock Dx Co., Ltd, Guangzhou, China) in mainland China, and (2) the FoundationOne®CDx assay (Foundation Medicine, Inc., Massachusetts, USA) in Taiwan and the global study (all patients recruited in Taiwan were part of the global population). The assays use the same biomarker rules, and good concordance has been established between them using commercial samples (unpublished data). Although the OncoScreen Plus assay does not detect *PTEN* rearrangements, the number of patients with tumor *PTEN* rearrangements in the Chinese cohort would likely be very low, given that in the global population this subset accounted for only two out of 594 patients with known biomarker results using the FoundationOne®CDx assay (unpublished data).

The safety profile of capivasertib–fulvestrant was manageable, and no new safety concerns versus the global population were identified; treatment discontinuation and dose reduction rates due to AEs were also infrequent and comparable to those in the global population[13]. Diarrhea, hyperglycemia and rash were the most commonly reported AEs in the capivasertib–fulvestrant group and were predominantly low grade. While higher rates of hyperglycemia AEs were reported in both treatment arms of the Chinese cohort versus the global population[13], grade ≥ 3 hyperglycemia was reported in only one patient (1.4%) in the capivasertib–fulvestrant group in the Chinese cohort. Moreover, the rates of hyperglycemia leading to treatment discontinuation or modification in the capivasertib–fulvestrant group were low and comparable between the Chinese cohort and the global population[13]. Higher rates of other laboratory-based AEs—such as anemia, hypokalemia, hypertriglyceridemia and decreased white

blood cell and neutrophil counts—were also reported in both treatment arms of the Chinese cohort versus the global population. The events were mostly low grade, and the rates of grade ≥ 3 events were similar between the Chinese cohort and the global population. There is a potential that investigators could be more likely to report laboratory value changes as AEs in the Chinese cohort due to local reporting habits, as noted previously in clinical studies in Chinese patients[40]. It is, however, difficult to conclude with certainty whether the increased frequency of laboratory value-based events was because of reporting habits, increased susceptibility in patients in the Chinese cohort, differences in baseline characteristics or other unknown factors.

Overall, the data from the Chinese cohort demonstrate that the benefit–risk profile of capivasertib–fulvestrant is positive and reflects the profile of the global population. Of note, the substantial PFS benefit observed in patients with NGS-confirmed *PIK3CA/AKT1/PTEN*-non-altered tumors warrants further exploration in this subpopulation of Chinese cohort patients, to identify potential characteristics associated with the PFS benefit exerted by capivasertib–fulvestrant. The combination of capivasertib–fulvestrant offers a potential treatment option for patients with HR-positive/HER2-negative advanced breast cancer that progressed on or after an endocrine-based regimen.

## Methods

### Study design and oversight

CAPItello-291 (ClinicalTrials.gov registration: NCT04305496) is a phase 3, double-blind, placebo-controlled, randomized study. Patients in the Chinese cohort were recruited in either the global phase 3 study or in an extended Chinese cohort with the same protocol and inclusion/exclusion criteria after recruitment to the global study had closed. The study was designed and overseen by an academic steering group that included representatives from AstraZeneca, the sponsor. An institutional review board and independent ethics committee reviewed the study protocol, amendments, and other relevant documents. A description of the protocol is in the Supplementary Information. Patients were recruited from 25 sites in mainland China and 3 NMPA-certified sites in Taiwan. The first and the last patient were enrolled in the study on 13 October 2020 and 3 January 2023, respectively.

### Human research participants

The study was also approved by the following local independent ethics committees (ECs) /institutional review boards (IRBs): China: Hubei Cancer Hospital, The Third Hospital of Nanchang, The First People's Hospital of Foshan, Affiliated Hospital of Hebei University, IRB EC of Linyi Cancer Hospital, The Second People's Hospital of Neijiang, IRB Shantou Central Hospital Ethics Committee; Taiwan: Research EC (REC) National Taiwan University, REC China Medical University Hospital, IRB Chang Gung Medical Foundation, IRB Taipei Veterans General Hospital, IRB National Cheng Kung University Hospital, IRB Koo Foundation Sun Yat-Sen Cancer Center, IRB E-DA Hospital, IRB Chi Mei Medical Center. The study was approved by the Human Genetics Resources Administration of China, and was conducted in accordance with the applicable International Council for Harmonization of Technical Requirements for Pharmaceuticals for Human Use and Good Clinical Practice guidelines and the principles of the Declaration of Helsinki. All patients gave written informed consent prior to enrollment.

### Study population

The study recruited pre-/peri- or postmenopausal women or men aged ≥ 18 years with breast cancer that was locally advanced (primary inoperable) or metastatic and was histologically confirmed to be HR-positive/HER2-negative, determined from the most recent tumor sample (primary or metastatic) per the American Society of Clinical Oncology and College of American Pathologists guideline

recommendations[41,42]. HR-positive was defined as estrogen receptor expression, with or without progesterone receptor expression. HER2-negative was defined as 0 or 1+ intensity on immunohistochemistry (IHC), or 2+ intensity on IHC and no amplification by in situ hybridization, or, if IHC was not done, no evidence of amplification by in situ hybridization.

Disease progression after prior aromatase inhibitor therapy with or without a CDK4/6 inhibitor was required, with disease progression defined as progression on a prior aromatase inhibitor in the metastatic setting or progression on or within 12 months of the end of treatment with a (neo)adjuvant aromatase inhibitor. Aromatase inhibitor therapy was not required to be the most recent treatment. Up to two prior endocrine therapy lines and one prior line of chemotherapy in the advanced setting were allowed. Patients with prior exposure to fulvestrant or another selective estrogen receptor degrader, or AKT, PI3K or mTOR inhibitors, were excluded, as were patients with diabetes requiring insulin or baseline glycated hemoglobin ≥ 8.0% (63.9 mmol/mol). Measurable disease (per Response Evaluation Criteria in Solid Tumors [RECIST] v1.1), or at least one lytic or mixed lytic–blastic bone lesion that could be assessed by computed tomography or magnetic resonance imaging, was required. Patients with an Eastern Cooperative Oncology Group/World Health Organization performance status of 0 or 1 were eligible.

Patients were required to provide tumor tissue for molecular analysis. Activating mutations in *PIK3CA* and *AKT1*, and inactivating alterations in *PTEN*, were determined centrally (post randomization) by NGS using the OncoScreen Plus assay in mainland China and the FoundationOne®CDx assay for patients enrolled in Taiwan. Both NGS assays use the same biomarker rules and have similar performance characteristics, although the OncoScreen Plus assay does not detect *PTEN* rearrangements. Good overall agreement has been established between the two assays using commercial samples (unpublished data). Patients whose tumors had at least one qualifying alteration in any of these three genes were defined as the population of patients with *PIK3CA/AKT1/PTEN*-altered tumors. Patients with tumors lacking a qualifying alteration detected in any of these three genes—or with an unknown NGS result—were included in the population of patients with *PIK3CA/AKT1/PTEN* non-altered tumors.

### Treatment

Patients were randomized 1:1 to receive oral capivasertib (400 mg twice daily for 4 days on followed by 3 days off) plus fulvestrant (500 mg intramuscularly given per standard-of-care every 14 days for the first three injections, then every 28 days), or matching placebo plus fulvestrant. The randomization scheme was produced using the AstraZeneca Global Randomisation system, which incorporates a standard procedure for generating random numbers. Blocked randomization was generated, and all centers used the same list to minimize imbalances in the number of patients assigned to each treatment group. Capivasertib and placebo were labeled using a unique kit identification number linked to the randomization scheme. To ensure treatment blinding, the film-coated tablets of capivasertib and placebo were identical in appearance and presented in the same packaging.

One cycle was defined as 4 weeks of capivasertib or placebo. Pre- or perimenopausal women also received a luteinizing hormone-releasing hormone agonist for the duration of the study treatment. Male patients could receive a concomitant luteinizing hormone-releasing hormone agonist, if deemed appropriate by the investigator. Randomization was stratified by region (Region 1: United States, Canada, Western Europe, Australia, and Israel; Region 2: Latin America, Eastern Europe and Russia; Region 3: Asia; for patients recruited as part of the global population only), the presence of liver metastases and prior use of CDK4/6 inhibitors.

Treatment continued until disease progression (per RECIST v1.1), unacceptable toxicity, withdrawal of consent or death. Dose reduction

of capivasertib or placebo was allowed from 400 mg to 320 mg and then to 200 mg, if required. Dose reductions or interruptions were made for grade ≥ 3 AEs attributed to capivasertib/placebo, or for lower grades as clinically appropriate, but dose reductions of fulvestrant were not allowed. Patients who discontinued one treatment for reasons other than disease progression could continue to receive the other.

### Endpoints

The dual primary endpoint was investigator-assessed PFS by RECIST v1.1 in the overall population and in patients with *PIK3CA/AKT1/PTEN*-altered tumors. Secondary endpoints included overall survival, PFS2, objective response rate, safety and tolerability.

### Procedures

Tumor assessments were performed per RECIST v1.1 by computed tomography and/or magnetic resonance imaging scans at screening (within 4 weeks before randomization), every 8 weeks for the first 18 months, and then every 12 weeks until disease progression. Radiographic bone scans were performed at screening and repeated as clinically indicated. Patients who discontinued study treatments for reasons other than disease progression continued to have scans every 8 weeks until disease progression (per RECIST v1.1). Patients entered the PFS2 follow-up period once they had discontinued study treatment due to progressive disease per RECIST v1.1. Progression on second-line treatment was assessed by the investigator 30 days after study treatment discontinuation, every 8 weeks for the first 2 years, and then every 12 weeks until second progression. Survival status and subsequent cancer therapies, following objective disease progression or treatment discontinuation, were documented 30 days after treatment discontinuation, every 8 weeks for the first 2 years, and then every 12 weeks until end of study, study withdrawal or death. AEs were recorded continuously until 30 days after treatment discontinuation and were graded using the National Cancer Institute (NCI) Common Terminology Criteria for Adverse Events (CTCAE) v5.0.

### Statistics

All statistical analyses were exploratory and performed only if sufficient numbers of events or patients were available (e.g. ≥ 20 PFS or overall survival events across both treatment groups), as prespecified in the statistical analysis plan. No formal sample size calculation was performed, no adjustment for multiplicity was performed and only descriptive statistics are presented. A cohort of approximately 134 randomized patients was planned, including patients recruited as part of the global study in mainland China and NMPA-certified sites in Taiwan. Sample size determination was conducted in collaboration with the China Regulatory Authority to ensure a sample size of Chinese participants that would be adequate to evaluate and demonstrate the consistency in the assessment of safety and efficacy of the capivasertib–fulvestrant combination between the global population and the Chinese cohort; considerations for consistency followed a previously published regional sample size allocation method[43].

Analysis of the dual primary endpoints was planned at approximately 77% maturity in both the overall Chinese cohort population (when 103 events of progression or death had occurred) and in the population of patients with *PIK3CA/AKT1/PTEN*-altered tumors (when approximately 41 events of progression or death had occurred, assuming a prevalence of 40%). Efficacy analyses included all the patients who underwent randomization. PFS was defined as the time from randomization until objective disease progression per RECIST v1.1 or death by any cause. The dual primary endpoints were tested with the use of a stratified log-rank test, with stratification according to previous use of a CDK4/6 inhibitor (yes or no) and the presence or absence of liver metastases (overall population only). Hazard ratios and associated two-sided 95% CIs were calculated from a stratified Cox

proportional hazards model fitted with the use of the PROC PHREG procedure in the SAS 9.4 software (SAS Institute, Cary, NC, USA), with the Efron method to control for ties. Kaplan-Meier survival plots are presented. Subgroup analyses were performed according to various factors (e.g. the presence of liver metastases or previous use of a CDK4/6 inhibitor) and are presented as forest plots. *Post hoc* exploratory PFS analysis was conducted in the extended Chinese cohort (i.e. excluding patients recruited in the global CAPItello-291 phase 3 study) using the same methodology and stratification factors as for the primary PFS analysis.

Overall survival was defined as the time from randomization until death by any cause. PFS2 was defined as the time from randomization to second progression (i.e. the earliest of either death or a progression event following treatment start after first progression). These secondary endpoints were analyzed similarly to the dual primary endpoints using a log-rank test and Cox proportional hazards model fitted with the use of the PROC PHREG procedure in the SAS 9.4 software, with the Efron method to control for ties. Kaplan-Meier survival plots are presented, and stratification factors are indicated in figure legends.

Objective response rate was defined as the percentage of patients with at least one complete or partial response per RECIST v1.1. The percentage of patients with an objective response was analyzed with the use of an unadjusted logistic regression model in patients with measurable disease.

Patients who received at least one dose of capivasertib, fulvestrant or placebo were included in the safety analyses. Safety data were summarized using descriptive statistics. The group term of rash was analyzed retrospectively, and comprised the preferred terms of rash, rash macular, rash maculopapular, rash papular and rash pruritic.

Change from baseline in the EORTC QLQ-C30 on-treatment scores was analyzed using a mixed effects model of all postbaseline scores for each visit. The model was analyzed using restricted maximum likelihood estimation and included treatment arm, visit, and treatment by visit interaction and stratification factors as explanatory variables, and the baseline score and baseline score by visit as covariates. Scaled residuals were examined to confirm the adequacy of the fitted model. The EORTC QLQ-C30 is assessed on a scale of 0 to 100, with higher scores indicating a higher quality of life. Summary statistics for each treatment arm were presented at timepoints where ≥20 patients in the respective treatment arm had an assessment and over one-third had received treatment. Time to deterioration was defined as the time from the date of randomization until the date of the first clinically meaningful deterioration (a sustained decrease of ≥ 10 points in the score from baseline). Overall compliance rates for each treatment group were calculated as the total number of patients with an evaluable questionnaire at baseline and at least one postbaseline timepoint, divided by the total number of patients expected to have completed at least a baseline questionnaire, multiplied by 100. Compliance over time was calculated separately for each visit (including baseline) as the number of patients with an evaluable questionnaire at the timepoint, divided by the number of patients still expected to complete questionnaires.

### Reporting summary

Further information on research design is available in the Nature Portfolio Reporting Summary linked to this article.

## Data availability

Source data is not available with this manuscript due to patient privacy, data underlying the findings and figures described in this manuscript may only be obtained in accordance with AstraZeneca's data sharing policy described at https://astrazenecagrouptrials.pharmacm.com/ST/Submission/Disclosure. Deidentified patient datasets (to General Data Protection Regulation standards, with link to patient code destroyed) would be available on request. Data could be

requested through Vivli at https://vivli.org/members/enquiries-about-studies-not-listed-on-the-vivli-platform/. AstraZeneca's Vivli member page is also available, outlining further details: https://vivli.org/ourmember/astrazeneca/. Some patients/countries may need to be excluded based on the informed consent form or country-level legislation (e.g. Chinese patients would be excluded based on Human Genetic Resources Regulations). Patients who have withdrawn consent for data use will also be removed from the shared dataset. Only clinical trial data may be shared. Patient-level image or genetic data are not available for access in our repository, in the interest of protecting patient privacy. Available documents include the clinical trial protocol, statistical analysis plan, informed consent form and clinical study report. Data can be available until the expiry of the Retention and Disposal Schedule of the data, based upon trial milestones at AstraZeneca. The data will be available upon approval of the request and signature of the Data Usage Agreement until, typically, one year starting on the date access was granted. Use of the data is restricted to the named users approved for the request and is made available to the requestor for one year from the date access was granted. Please refer to the Data Usage Agreement (https://vivli.org/resources/vivli-data-use-agreement/; non-negotiable contract for data accessors) for more information.

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

## Acknowledgements

The authors thank the patients and their families who participated in the CAPItello-291 study, as well as the investigators, co-investigators, study staff and the Steering Committee. This study was sponsored by AstraZeneca, and was designed and overseen by an academic steering group that included representatives from AstraZeneca. Capivasertib was discovered by AstraZeneca subsequent to a collaboration with Astex Therapeutics (and its collaboration with the Institute of Cancer Research and Cancer Research Technology Limited). Medical writing support, under the direction of the authors, was provided by Stavroula Bitsi, PhD, of BOLDSCIENCE Ltd, funded by AstraZeneca in accordance with Good Publication Practice guidelines 2022.

## Author contributions

X.H. and N.C.T. served as principal investigators, participated in the Steering Committee and contributed to the study design. X.H., Q.Z., T.S., H.X., W.L., Y.T., Y-S.L., L-M.T., M.Y., H.L., D.P., S-C.C., W.C., O.J., J.W., X. Wu, Xian Wang, A.Z., Xiaojia Wang, and N.C.T. recruited and/or treated patients. X.H., J.M.C., E.F., L.J., X.Z., and N.C.T. contributed to data analysis and interpretation. All authors contributed to the critical review and revision of the manuscript and had final responsibility for the decision to submit it for publication.

## Competing interests

X.H. has participated on advisory boards for AstraZeneca, received institutional funding for an investigator-initiated trial from Merck Sharpe and Dohme, and has served as a local Principal Investigator for AstraZeneca and Novartis. Y-S.L. has participated on advisory boards for Novartis, Eli Lilly, Merck Sharpe and Dohme, Roche, Pfizer, AstraZeneca, Eisai, Daiichi Sankyo, and Gilead Sciences, has been an invited speaker for Novartis, Eli Lilly, Merck Sharpe and Dohme, Roche, Pfizer, AstraZeneca, Eisai, Daiichi Sankyo, and Gilead Sciences, has served as a local Principal Investigator for Novartis, Roche, Pfizer, Merck Sharpe and Dohme, Eli Lilly, Eisai, AstraZeneca, Gilead Sciences and Jellox, has served as a chair/co-chair of a clinical trial steering committee for Novartis, and has held an uncompensated advisory role for AstraZeneca. J.M.C., E.F., L.J., and X.Z. are full-time employees of AstraZeneca. N.C.T. has participated in advisory boards AstraZeneca, Eli Lilly, Exact Sciences, Gilead Sciences, GlaxoSmithKline, Guardant Health, Inivata, Novartis, Pfizer, Roche/Genentech, Relay Therapeutics, Repare Therapeutics and Zentalis Pharmaceuticals, has received institutional research funding from AstraZeneca, Inivata, Invitae, Merck Sharpe and Dohme, Natera, Personalis, Pfizer and Roche/Genentech, and has received research assays/materials from Guardant Health and Bio-Rad. Q.Z., T.S., H.X., W.L., Y.T., L-M.T., M.Y., H.L., D.P., S-C.C., W.C., O.J., J.W., X. Wu, Xian Wang, A.Z. and Xiaojia Wang declare no competing interests.

## Additional information

Xichun Hu [1,23] ✉, Qingyuan Zhang [2,23], Tao Sun [3], Huihua Xiong[4], Wei Li[5], Yuee Teng[6], Yen-Shen Lu [7], Ling-Ming Tseng[8], Min Yan [9], Hongsheng Li[10], Danmei Pang[11], Shin-Cheh -Chen [12], Wenyan Chen[13], Ou Jiang[14], Jingfen Wang[15], Xinhong Wu[16], Xian Wang[17], Aimin Zang[18], Xiaojia Wang [19], Julie M. Collins[20], Ethan Fan [21], Lin Jiang[21], Xiaoling Zeng[21] & Nicholas C. Turner [22]

[1]Department of Medical Oncology, Fudan University Shanghai Cancer Center; Fudan University, Shanghai, China. [2]Harbin Medical University Cancer Hospital, Harbin, China. [3]Liaoning Cancer Hospital, Shenyang, China. [4]Tongji Hospital, Tongji Medical College, Huazhong University of Science & Technology, Wuhan, China. [5]The First Hospital of Jilin University Cancer Center, Changchun, China. [6]The First Hospital of China Medical University, Shenyang, China. [7]National Taiwan University Hospital, Taipei, Taiwan. [8]Taipei Veterans General Hospital, Taipei, Taiwan. [9]The Affiliated Cancer Hospital of Zhengzhou University, Zhengzhou, China. [10]Affiliated Tumor Hospital of Guangzhou Medical University, Guang Zhou Shi, China. [11]The First People's Hospital of Foshan, Foshan, China. [12]Division of Breast Surgery, Chang Gung Memorial Hospital, Linkou, Taoyuan, Taiwan. [13]The Third Hospital of Nanchang, Nanchang, China. [14]The Second People's Hospital of Neijiang, Neijiang, China. [15]Linyi Cancer Hospital, Linyi City, China. [16]Department of Breast Surgery, Hubei Cancer Hospital, Tongji Medical College, Wuhan, Hubei, China. [17]Sir Run Run Shaw Hospital, Zhejiang University School of Medicine, Hangzhou, China. [18]Affiliated Hospital of Hebei University, Baoding, China. [19]Zhejiang Cancer Hospital, Hangzhou, China. [20]Oncology R&D, AstraZeneca, Gaithersburg, MD, USA. [21]Oncology R&D, AstraZeneca, Shanghai, China. [22]Royal Marsden Hospital, Institute of Cancer Research, London, UK. [23]These authors contributed equally: Xichun Hu, Qingyuan Zhang ✉e-mail: xchu2009@hotmail.com

