## [Transparent Peer Review file · Nature Communications]

Capivasertib plus fulvestrant in patients with HR-positive/HER2-negative advanced breast cancer: phase 3 CAPitello-291 study extended Chinese cohort

Corresponding Author: Professor Xichun Hu

Version 1:

Reviewer comments:

Reviewer #1

(Remarks to the Author)

The authors have addressed some concerns in the revised manuscript. However, concern remains regarding the main novelty of the manuscript: numerically longer PFS benefit in Chinese patients with AKT pathway non-altered tumors than the global population.

Although the authors said the results are hypothesis-generating, warranting further translational research and clinical exploration, no clear hypothesis or future research was discussed in the manuscript.

Reviewer #2

(Remarks to the Author)

For such subset and extended cohort analyses following the main publication, the key questions of interest would be whether there are any ethnic differences in characteristics, efficacy and tolerability.

The manuscript is generally well written, but the scientific content could be further improved for publication in a Nature journal.

The most noteworthy result to me is the greater PFS benefit of adding capivasertib in the non-altered cohort compared to the Capitello-291 global population. While this is not the primary population of interest, it would be important to understand why there may be a potential benefit of adding capivasertib in patients with non-altered tumors, or in all patients (which is the label for capivasertib in Australia).

Since the next generation assays used should have covered genes involving receptor tyrosine kinase such as HER and FGFR family that can activate Akt downstream, as mentioned in the discussion, description of such co-alterations would be insightful, even in the absence of statistical comparison.

The other "novel" finding that struck me is the higher frequency of hyperglycaemia compared to the global cohort in Capitello-291. Do the authors, or team analysing/interpreting the data think that this is a true difference? More could be discussed regarding this.

Reviewer #3

(Remarks to the Author)

Thank you for your revisions and rebuttals.

For comment 1, I appreciated the addition of results for the n=110, which would also be appreciated by anyone performing a systematic review, particularly if they intend meta-analysis or meta-regression. While I would prefer all analyses to be

repeated using the n=110, I think that could be unwieldy and I would hope, I that you would assist anyone wanting to look at the other outcomes for the n=110 on request. I think readers would appreciate some signposting of these results. For example, around Line 110, you could state that unplanned (or post hoc as you use on Line 158) exploratory analyses also examined the new Chinese cohort (n=110) only.

For comment 2, I'm sorry if this seems like a creeping request, but could you extend "to ensure adequate participation to satisfy China Regulatory Authority requirements" so that the reader can better understand exactly what these requirements entailed? For example "...to ensure adequate participation to satisfy China Regulatory Authority requirements that ??????" on Line 433. 134 seems a slightly unusual requirement so I am presuming that this must have results from some expectation, e.g., so that with three-quarters recording an event, there would be at least 100 events expected (c.f., the 102 on Line 130). That would be an arbitrary rule, but less mysterious than n=134 is for me at the moment.

Related to this, for comment 5, I still think regularization would be the better strategy (your rebuttal didn't directly address this point). Peduzzi, et. al., would have recommended at least 20 events AND at least 20 non-events in the analytic sample for a particular logistic or Cox's proportional hazards regression analysis if there were exactly two continuous/binary/dummy predictor variables (10 events [and 10 non-events] per variable). By itself, the reference wouldn't appear to explain the 'rule' used here to the reader. I'm still confused myself as to the basis for the rule of at least 20 events (why not 10, 25, or 30, as examples?) Can you explain where this rule was obtained in the manuscript for the curious reader? Related to this, why the at least 20 for the summary statistics (Line 478)? Peduzzi only speaks about logistic and Cox's proportional hazards regression in their two articles on this topic.

For comment 4, I couldn't see any additions that addressed model checks or diagnostics. I'm not expecting a lengthy explanation of this, but some comments would be reassuring to the reader, I believe.

As a very minor point, my comment 3, about the name "mixed model repeated measures analysis" (Line 473), was about it being potentially confused by readers as a "Repeated Measured ANOVA" (which I have seen mis-described using similar wording to that in your manuscript, which is not a failing of your work but rather a confusion created by others). I'll leave this to the authors, but phrasing such as "a linear mixed model was used to analyses all postbaseline scores from each visit" seems clear and unambiguous to me. The same would apply to Lines 861–862.

As a last minor point, for comment 6, it seems to me that small details like using REML (restricted or residual maximum likelihood) could easily enough be added to the manuscript without overwhelming the reader (e.g., as parenthetical comments).

Version 2:

Reviewer comments:

Reviewer #4

(Remarks to the Author)

The main finding of this study is prolonged PFS in the overall Chinese population and in certain subgroups. The patients analyzed in this study is part of the phase III CAPItello-291 trial. It is not clear if this Chinese cohort have well balanced baseline characteristics between the treatment groups and no statistical techniques are employed to deal with potential biases due to the break of randomization. The reported HRs for PFS might be biased as a result. This is a major flaw and needs to be addressed using propensity score based approaches.

Comment 1: the hazard ratio is reported as non-calculable for certain subgroups. It will be more informative to reader if the number of events in these subgroups are reported in the main text rather than the supplementary materials. Suggest rephrase to state: we choose not to calculate hazard ratios for these subgroups due to the smaller number of events.

line 148: The hazard ratio was non-calculable for patients with an unknown NGS result due to an insufficient number of events (Supplementary Fig. 2c).

line 154: Benefit was also seen in clinically relevant subgroups of patients with PIK3CA/AKT1/PTEN-altered tumors, but the 95% CIs were wider and the hazard ratios non-calculable in some instances

Comment 2: line 167, objective response rate should be reported together with 95% CIs.

Comment 3: line 198 should be rephrased to state median OS is not reached.

Comment 4: line 497 mixed model repeated measures analysis is not correct statistical terminology. It should be corrected to mixed effects model and patient-level random effects to account for correlations between different visits.

We are grateful to the peer reviewers for providing their comments. Our responses are described below point-by-point. We hope that the incorporated revisions address the comments in a satisfactory manner.

Reviewer 1 comments	Author response	Changes made	Page number in revised
1. The submitted manuscript is an extended study of CAPItello-291 randomized phase 3 study (NCT04305496) which was already published last year (N Engl J Med 2023;388:2058-70). CAPItello-291 study demonstrated the combination of capivasertib–fulvestrant therapy resulted in better progression-free survival than fulvestrant alone in previously AI-treated advanced HR+ breast cancer. The current study comprised 24 patients from the global study and 110 more Chinese patients with the same protocol. The PFS in overall population were similar between two studies (median PFS: 6.9 vs 2.8 months in the current study, 7.2 vs 3.6 months in CAPItello-291) as well as the PFS in patients with PIK3CA/AKT1/PTEN-altered tumors (5.7 vs 1.9 months in the current study, 7.3 vs 3.1 months in CAPItello-291 study). The PFS improvement in both overall population and PIK3CA/AKT1/PTEN-altered tumors reflected the versatility and complexity of the role of the AKT pathway in cancer progression. Interestingly, the PFS in patients with AKT pathway non-altered tumors in the extended study was numerically better in the Chinese population than the global population (9.2 vs 2.7 months in the current study, 5.3 vs 3.7 months in CAPItello-291 study). Nevertheless, it is difficult to draw any conclusion from the data. More genetic information and biomarker analysis are needed. The adverse event profiles were similar between two studies, too. Thus, this extended study only generated limited new information beyond the global study.	We thank the reviewer for taking the time to review our manuscript and for providing their input. We report, for the first time, data for the combination of capivasertib–fulvestrant in a cohort of Chinese patients, the majority of whom were not included in the global study. We believe that our results are hypothesis-generating, warranting further translational research and clinical exploration. Of note, capivasertib–fulvestrant was recently approved in Australia as an indication for patients with advanced breast cancer irrespective of PIK3CA/AKT1/PTEN alteration status. Unfortunately, additional biomarker analysis is not available for inclusion in this manuscript because of country-specific regulatory restrictions.	–	–

Reviewer 1 comments	Author response	Changes made	Page number in revised
2. Minor points 2a. The investigators should include a statement of ethics committee since this is an extended, multicenter study.	The study received approval from the following independent ethics committees/institutional review boards: China: Hubei Cancer Hospital, The Third Hospital of Nanchang, The First People's Hospital of Foshan, Affiliated Hospital of Hebei University, IRB EC of Linyi Cancer Hospital, The Second People's Hospital of Neijiang, IRB Shantou Central Hospital Ethics Committee; Taiwan: REC National Taiwan University, REC China Medical University Hospital, IRB Chang Gung Medical Foundation, IRB Taipei Veterans General Hospital, IRB National Cheng Kung University Hospital, IRB Koo Foundation Sun Yat-Sen Cancer Center, EDH IRB E-DA Hospital, IRB Chi Mei Medical Center.	The following has been added to the manuscript: "The study was also approved by the following local independent ethics committees (ECs) /institutional review boards (IRBs): China: Hubei Cancer Hospital, The Third Hospital of Nanchang, The First People's Hospital of Foshan, Affiliated Hospital of Hebei University, IRB EC of Linyi Cancer Hospital, The Second People's Hospital of Neijiang, IRB Shantou Central Hospital Ethics Committee; Taiwan: Research EC (REC) National Taiwan University, REC China Medical University Hospital, IRB Chang Gung Medical Foundation, IRB Taipei Veterans General Hospital, IRB National Cheng Kung University Hospital, IRB Koo Foundation Sun Yat-Sen Cancer Center, IRB E-DA Hospital, IRB Chi Mei Medical Center."	16

Reviewer 1 comments	Author response	Changes made	Page number in revised
2b. The rationale of the sample size was not given for an interventional clinical trial.	The sample size was chosen to ensure an adequate number of Chinese patients were randomized to meet China Regulatory Authority requirements. To note, the selection also abides by the International Council for Harmonisation of Technical Requirements for Pharmaceuticals for Human Use (ICH) E17 Guideline on General Principles for Planning and Design of Multi-Regional Clinical Trials. We mention that the study “was conducted in accordance with the applicable International Council for Harmonisation of Technical Requirements for Pharmaceuticals for Human Use and Good Clinical Practice guidelines” on page 15. Information has now been added to the manuscript to also satisfy the comment by Reviewer 3 “What was the basis for the n=134 required? I couldn’t see a specific justification for it. In the SAP, you say ‘This is to ensure adequate participation of Chinese patients to satisfy China Regulatory Authority requirements.’ If this is the only basis for the 134, I think that this needs to be included in the manuscript itself.”	We have clarified that (added text in bold): “A cohort of approximately 134 randomized patients was planned, including patients recruited as part of the global study in mainland China and NMPA-certified sites in Taiwan, to ensure adequate participation to satisfy China Regulatory Authority requirements”	20

Reviewer 2 comments	Author response	Changes made	Page number in revised
1. The manuscript is generally well written in the style of medical writers. No concerns with the presentation of the methods, results and discussion. However, such a subset analysis, even though supplemented by an extended cohort, lacks novelty, and sample size is also relatively restricted.	We thank the reviewer for taking the time to review our manuscript and for raising no concerns regarding the presentation of the methods, results and discussion. We report, for the first time, data for the combination of capivasertib–fulvestrant in a cohort of Chinese patients, the majority of whom were not included in the global study; thus, our analysis does not represent a subset analysis. We believe that our results are hypothesis-generating, warranting further translational research and clinical exploration, particularly in Chinese patients. The sample size was chosen to ensure an adequate number of Chinese patients were randomized to meet China Regulatory Authority requirements.	–	–

Reviewer 3 comments	Author response	Changes made	Page number in revised
1. I am a biostatistician and have focused on those aspects when considering this interesting study, which seems to have been well-conducted and is generally well-presented, looking at both from a biostatistical perspective. I have only a few comments, although I think that these are likely to involve significant work to address. The original participants from the Chinese arm appear included in https://doi.org/10.1056/NEJMoa2214131 (there “Between June 2, 2020, and October 13, 2021, a total of 901 patients were enrolled at 193 centers in 19 countries”) with this augmented by the additional 110 patients to achieve n=134 in order to satisfy China Regulatory Authority requirements. The analysis set here is based on patients enrolled “Between 13 October 2020 and 3 January 2023, 134 patients (24 patients from the global population and 110 patients from an extended cohort) were recruited from 25 sites in mainland China.” If I am correct in reading that 24 participants are included in both analyses, this presents some difficulties for me. An immediate one would be the difficulty of using both sets of results within a meta-analysis since their samples appear to overlap. At the same time, I appreciate the regulatory requirements and benefits of using all n=134 here. As a compromise position, I wonder if at least a secondary set of analyses could be provided using only the 110 additional patients added from the main trial in order to enable systematic reviews and meta-analyses to combine results from the global study and the new data presented here.	We thank the reviewer for their review of our manuscript and their valuable insights based on their experience as a biostatistician. We appreciate the rationale behind their request and have, therefore, conducted post hoc progression-free survival analysis for the patients recruited as part of the extended study only. The results for the extended cohort were in line with those in the overall Chinese cohort. We hope the additional analysis provided satisfies the request by peer review and will facilitate potential future meta-analyses.	Relevant additions have been incorporated into the Results, Discussion and Methods sections.	8, 12, 21, 55, 56

Reviewer 3 comments	Author response	Changes made	Page number in revised
2. What was the basis for the n=134 required? I couldn't see a specific justification for it. In the SAP, you say "This is to ensure adequate participation of Chinese patients to satisfy China Regulatory Authority requirements." If this is the only basis for the 134, I think that this needs to be included in the manuscript itself.	The reviewer is correct, the sample size was chosen to ensure an adequate number of Chinese patients were randomized to meet China Regulatory Authority requirements. To note, the selection also abides by the International Council for Harmonisation of Technical Requirements for Pharmaceuticals for Human Use (ICH) E17 Guideline on General Principles for Planning and Design of Multi-Regional Clinical Trials. We mention that the study "was conducted in accordance with the applicable International Council for Harmonisation of Technical Requirements for Pharmaceuticals for Human Use and Good Clinical Practice guidelines" on page 15.	We have clarified that (added text in bold): "A cohort of approximately 134 randomized patients was planned, including patients recruited as part of the global study in mainland China and NMPA-certified sites in Taiwan, to ensure adequate participation to satisfy China Regulatory Authority requirements".	20

3. It appears that “Randomization was stratified by region (for patients recruited as part of the global population only), the presence of liver metastases and prior use of CDK4/6 inhibitors.” If region here refers to regions within China, as seems likely, this would seem to lead to all three variables needing to be included in the analysis (see https://doi.org/10.1002/sim.4431 for an explanation of the problems with not adjusting for stratification and minimisation variables). I see that stratification for two of these was used for the log-rank tests but not region. It was unclear whether the logistic regression models (Line 403) or the linear mixed models (Line 409, and I recommend against the description used there as it seems to sometimes be confused with a RM-ANOVA) adjusted for any of the stratification variables. Personally, I would be inclined to treat region as a random effect.	1. We appreciate that “region” may have not been adequately defined in the manuscript. Therefore, we would like to clarify that “region” does not refer to regions within China, but geographical locations relevant to the global study. Therefore, region as a stratification factor does not apply for most of the patients included in the Chinese cohort. 2. Regarding the analysis for objective response (line 403 of initial submission), it is indicated that “The percentage of patients with an objective response was analyzed with the use of an unadjusted logistic regression model in patients with measurable disease” Stratification variables used for each analysis are indicated in figure legends. 3. Regarding the analysis for the change from baseline in the EORTC QLQ-C30 on-treatment scores (line 409 of initial submission), the model included treatment arm, visit, and treatment by visit interaction and stratification factors as explanatory variables, and the baseline score and baseline score by visit as covariates. Data were derived using MMRM (mixed model repeated measures), so we believe this should remain as is in the manuscript.	We have clarified that (added text in bold): 1. “Randomization was stratified by region (Region 1: United States, Canada, Western Europe, Australia, and Israel; Region 2: Latin America, Eastern Europe and Russia; Region 3: Asia; for patients recruited as part of the global population only) [...]”. 2. – 3. “Change from baseline in the EORTC QLQ-C30 on-treatment scores was analyzed using a mixed model repeated measures analysis of all postbaseline scores for each visit. The model included treatment arm, visit, and treatment by visit interaction and stratification factors as explanatory variables, and the baseline score and baseline score by visit as covariates”. This text has also been added to the legend of Extended Fig. 5.	1. 19 2. – 3. 22, 58
---	---	--	---

Reviewer 3 comments	Author response	Changes made	Page number in revised
	We do not include any mention to ANOVA throughout.		
4. For the linear mixed models, why wasn't baseline value included as a covariate? This usually increases power. A brief description of model diagnostics used in the manuscript itself would be helpful and reassuring.	The model included treatment arm, visit, and treatment by visit interaction and stratification factors as explanatory variables, and the baseline score and baseline score by visit as covariates, as mentioned above.	We have added that: "Change from baseline in the EORTC QLQ-C30 on-treatment scores was analyzed using a mixed model repeated measurements analysis of all postbaseline scores for each visit. The model included treatment arm, visit, and treatment by visit interaction and stratification factors as explanatory variables, and the baseline score and baseline score by visit as covariates". This text has also been added to the legend of Extended Fig. 5.	22, 58

Reviewer 3 comments	Author response	Changes made	Page number in revised
5. The 'non-calculable' results with < 20 events (e.g., Line 385) seems an unnecessary limitation. Exact or (my preference) penalized Cox or logistic regression would be options here, or Bayesian approaches could be used for regularisation. The requirement of 20 events didn't appear to be justified and so I'm guessing it might be based on Peduzzi, et al.?	We thank the reviewer for their suggestion. All analyses were conducted per protocol. In cases where there were <20 events across treatment groups, we do not present the analysis to avoid overinterpretation based on limited data. We have added the supporting reference by Peduzzi et al. 1996 to the manuscript.	The reference by Peduzzi et al. 1996 has been added to the following sentence: "All statistical analyses were exploratory and performed only if sufficient numbers of events or patients were available (e.g. ≥ 20 PFS or overall survival events across both treatment groups)".	20
6. The code used for analyses would be a useful supplement to answer specific questions around analyses (e.g., was REML used for the linear mixed models?) if it was felt difficult or distracting to include all of these in the text. I was unable to find answers to some of my biostatistical queries even in the SAP and code has the virtue of allowing one to answer these oneself.	We indeed believe that adding further analysis details to the manuscript would be distracting and overly complicated for the majority of the target readership. We confirm that REML (restricted maximum likelihood estimation) was used for the linear mixed models. Details on the analysis, including sample code, can be found in the SAP on pages 87–88. We are happy to clarify further queries regarding analyses but believe that providing an extensive supplement on code would be out of scope for this manuscript.	–	–

Reviewer 1 comments	Author response	Changes made
1. The authors have addressed some concerns in the revised manuscript. However, concern remains regarding the main novelty of the manuscript: numerically longer PFS benefit in Chinese patients with AKT pathway non-altered tumors than the global population. Although the authors said the results are hypothesis-generating, warranting further translational research and clinical exploration, no clear hypothesis or future research was discussed in the manuscript.	We thank the reviewer for taking the time to review the manuscript and for their insightful comments. Unfortunately, as mentioned in our previous response to the reviewer, additional biomarker analysis is not available for inclusion in this manuscript because of country-specific regulatory restrictions. We have, however, expanded the discussion on the topic to in response to the reviewer's comment.	Addition to the Discussion: “Additionally, exploratory biomarker analyses in the neoadjuvant triple-negative breast cancer setting showed that high levels of phosphorylated AKT1 in the absence of PIK3CA/AKT/PTEN alterations was associated with other pathway activation mechanisms, including deficiency of PTEN protein expression and enhanced activation of upstream receptor tyrosine kinases. Finally, crosstalk between the AKT and estrogen receptor pathways can reciprocally compensate monotherapy treatment in certain tumors, indicating that simultaneous inhibition of both pathways could have an impact in the overall population independent of alteration status. Hence, further research in Chinese patients with PIK3CA/AKT1/PTEN-non-altered tumors could be conducted to determine mechanisms, apart from alterations in PIK3CA/AKT1/PTEN, which could be associated with the PFS benefit exerted by capivasertib–fulvestrant.”

Reviewer 2 comments	Author response	Changes made
General comment: For such subset and extended cohort analyses following the main publication, the key questions of interest would be whether there are any ethnic differences in characteristics, efficacy and tolerability. The manuscript is generally well written, but the scientific content could be further improved for publication in a Nature journal.		
1. The most noteworthy result to me is the greater PFS benefit of adding capivasertib in the non-altered cohort compared to the Capitelto-291 global population. While this is not the primary population of interest, it would be important to understand why there may be a potential benefit of adding capivasertib in patients with non-altered tumors, or in all patients (which is the label for capivasertib in Australia). Since the next generation assays used should have covered genes involving receptor tyrosine kinase such as HER and FGFR family that can activate Akt downstream, as mentioned in the discussion, description of such co-alterations would be insightful, even in the absence of statistical comparison.	We thank the reviewer for taking the time to review the revised manuscript and for their insightful comments. We agree that the potential benefit of capivasertib plus fulvestrant in patients with PIK3CA/AKT1/PTEN-non-altered tumors is worthy of further investigation. Unfortunately, additional biomarker analysis is not available for inclusion in this manuscript because of country-specific regulatory restrictions. We have, however, expanded the discussion on the topic in response to the reviewer's comment.	Addition to the Discussion: “Additionally, exploratory biomarker analyses in the neoadjuvant triple-negative breast cancer setting showed that high levels of phosphorylated AKT1 in the absence of PIK3CA/AKT1/PTEN alterations was associated with other pathway activation mechanisms, including deficiency of PTEN protein expression and enhanced activation of upstream receptor tyrosine kinases. Finally, crosstalk between the AKT and estrogen receptor pathways can reciprocally compensate monotherapy treatment in certain tumors, indicating that simultaneous inhibition of both pathways could have an impact in the overall population independent of alteration status. Hence, further research in Chinese patients with PIK3CA/AKT1/PTEN-non-altered tumors could be conducted to determine mechanisms, apart from alterations in PIK3CA/AKT1/PTEN, which could be associated with the PFS benefit exerted by capivasertib–fulvestrant.”

2. The other "novel" finding that struck me is the higher frequency of hyperglycaemia compared to the global cohort in Capitello-291. Do the authors, or team analysing/interpreting the data think that this is a true difference? More could be discussed regarding this.	We have added the relevant data for dose modifications/discontinuation of capivasertib/placebo to the Results and have expanded the relevant section in the Discussion to indicate that local reporting habits could also play a role.	Addition to the Results: "Hyperglycemia AEs led to discontinuation of capivasertib/placebo in one patient (1.4%) receiving capivasertib and no patients receiving placebo, and to dose interruption in two patients (2.8%) receiving capivasertib and no patients receiving placebo. No dose reduction of capivasertib/placebo was required due to hyperglycemia AEs." Additions to the Discussion (in bold): "While higher rates of hyperglycemia AEs were reported in both treatment arms of the Chinese cohort versus the global population¹³, grade ≥ 3 hyperglycemia was reported in only one patient (1.4%) in the capivasertib–fulvestrant group in the Chinese cohort. Moreover, the rates of hyperglycemia leading to treatment discontinuation or modification in the capivasertib–fulvestrant group were low and comparable between the Chinese cohort and the global population. Higher rates of other laboratory-based AEs—such as anemia, hypokalemia, hypertriglyceridemia and decreased white blood cell and neutrophil counts—were also reported in both treatment arms of the Chinese cohort versus the global population. The events were mostly low grade, and the rates of grade ≥ 3 events were similar between the Chinese cohort and the global population. There is a potential that investigators could be more likely to report laboratory value changes as adverse events in the Chinese cohort due to local reporting habits, as noted previously in
--	---	---

Reviewer 2 comments	Author response	Changes made
		clinical studies in Chinese patients. It is, however , difficult to conclude with certainty whether the increased frequency of laboratory value-based events was because of reporting habits , increased susceptibility in patients in the Chinese cohort, differences in baseline characteristics or other unknown factors.

Reviewer 3 comments	Author response	Changes made
1. Thank you for your revisions and rebuttals. For comment 1, I appreciated the addition of results for the n=110, which would also be appreciated by anyone performing a systematic review, particularly if they intend meta-analysis or meta-regression. While I would prefer all analyses to be repeated using the n=110, I think that could be unwieldy and I would hope that you would assist anyone wanting to look at the other outcomes for the n=110 on request. I think readers would appreciate some signposting of these results. For example, around Line 110, you could state that unplanned (or post hoc as you use on Line 158) exploratory analyses also examined the new Chinese cohort (n=110) only.	We thank the reviewer for taking the time to review the revised manuscript and for their insightful comments. We have amended the text as suggested.	Addition to the Results in bold: “Between 13 October 2020 and 3 January 2023, 134 patients (24 patients from the global population and 110 patients from an extended cohort) were recruited from 25 sites in mainland China (n = 118) and 3 National Medical Products Administration (NMPA)-certified sites in Taiwan (n = 16). Patients were randomized to capivasertib–fulvestrant (n = 71) or placebo–fulvestrant (n = 63; Supplementary Fig. 1). Results are reported for the overall cohort of 134 patients, but exploratory PFS analysis was also conducted in the extended cohort of 110 patients.”

2. For comment 2, I'm sorry if this seems like a creeping request, but could you extend "to ensure adequate participation to satisfy China Regulatory Authority requirements" so that the reader can better understand exactly what these requirements entailed? For example "...to ensure adequate participation to satisfy China Regulatory Authority requirements that ????????" on Line 433. 134 seems a slightly unusual requirement so I am presuming that this must have results from some expectation, e.g., so that with three-quarters recording an event, there would be at least 100 events expected (c.f., the 102 on Line 130). That would be an arbitrary rule, but less mysterious that n=134 is for me at the moment.	We have added additional clarifications. Determination of the sample size was conducted in collaboration with the China Regulatory Authority to ensure consistency in the assessment of the combination treatment between the global population and the Chinese cohort. Detailed definition of consistency can be found in the previously published regional sample size allocation method used (PMDA guideline [Notification No. 0928010] 2007 - "Basic Principles on Global Clinical Trials."). As mentioned in the SAP, "The PFS Primary Analysis will take place after PFS reaches approximately 77% maturity (542 events) in the overall population and approximately 77% maturity in patients whose tumors harbor an eligible PIK3CA/AKT1/PTEN alteration, based on a prevalence of ~40-45% (and 174 events will have been observed if a test failure rate is 20%)." As stated in the primary global CAPItello-291 publication (Turner et al. N Engl J Med 2023;388:2058–2070), the primary analysis for the global cohort was conducted when 551 events of disease progression or death had occurred in the overall population (551/708; 77.8% maturity) and 236 events had occurred in the altered population (236/289; 81.7% maturity). The China SAP addendum indicates that efficacy analyses for the Chinese cohort would be performed when the PFS data from the Chinese cohort would be of similar maturity to	Amends/additions to the Methods in bold: "A cohort of approximately 134 randomized patients was planned, including patients recruited as part of the global study in mainland China and NMPA-certified sites in Taiwan. Sample size determination was conducted in collaboration with the China Regulatory Authority to ensure a sample size of Chinese participants that would be adequate to evaluate and demonstrate the consistency in the assessment of safety and efficacy of the capivasertib–fulvestrant combination between the global population and the Chinese cohort; considerations for consistency followed a previously published regional sample size allocation method.
---	---	---

Reviewer 3 comments	Author response	Changes made
	those in the global cohort. We already mention this in the Methods as follows: “Analysis of the dual primary endpoints was planned at approximately 77% maturity in both the overall Chinese cohort population (when 103 events of progression or death had occurred) and in the population of patients with PIK3CA/AKT1/PTEN-altered tumors (when approximately 41 events of progression or death had occurred, assuming a prevalence of 40%).”	

Reviewer 3 comments	Author response	Changes made
3. Related to this, for comment 5, I still think regularization would be the better strategy (your rebuttal didn't directly address this point). Peduzzi, et. al., would have recommended at least 20 events AND at least 20 non-events in the analytic sample for a particular logistic or Cox's proportional hazards regression analysis if there were exactly two continuous/binary/dummy predictor variables (10 events [and 10 non-events] per variable). By itself, the reference wouldn't appear to explain the 'rule' used here to the reader. I'm still confused myself as to the basis for the rule of at least 20 events (why not 10, 25, or 30, as examples?) Can you explain where this rule was obtained in the manuscript for the curious reader? Related to this, why the at least 20 for the summary statistics (Line 478)? Peduzzi only speaks about logistic and Cox's proportional hazards regression in their two articles on this topic.	As stated in our previous response, all analyses were conducted per the statistical analysis protocol and we have now stated this in the manuscript. This is standard practice for AstraZeneca-sponsored studies to avoid overinterpretation based on limited data, with our thinking derived from the Peduzzi et al. reference and further adjusted by our statistical team. We have removed the reference to avoid confusion that it fully supports our rationale. The same rule was applied for the primary CAPItello-291 publication (Turner et al. N Engl J Med 2023;388:2058–2070). We would propose that no further analysis of limited data that is not pre-specified in the protocol is carried out.	Additions to the Methods in bold: “All statistical analyses were exploratory and performed only if sufficient numbers of events or patients were available (e.g. ≥ 20 PFS or overall survival events across both treatment groups), as prespecified in the statistical analysis plan.”

Reviewer 3 comments	Author response	Changes made
4. For comment 4, I couldn't see any additions that addressed model checks or diagnostics. I'm not expecting a lengthy explanation of this, but some comments would be reassuring to the reader, I believe.	In response to the reviewer's comment, we previously added the following: "The model included treatment arm, visit, and treatment by visit interaction and stratification factors as explanatory variables, and the baseline score and baseline score by visit as covariates". As indicated, baseline score and baseline score by visit were included as covariates. The model is consistent with that pre-specified in the statistical analysis plan and so no formal model building was performed.	–
5. As a very minor point, my comment 3, about the name "mixed model repeated measures analysis" (Line 473), was about it being potentially confused by readers as a "Repeated Measured ANOVA" (which I have seen mis-described using similar wording to that in your manuscript, which is not a failing of your work but rather a confusion created by others). I'll leave this to the authors, but phrasing such as "a linear mixed model was used to analyses all postbaseline scores from each visit" seems clear and unambiguous to me. The same would apply to Lines 861–862.	We thank the reviewer for their comment. However, we would prefer to retain the term for "mixed model repeated measures analysis", as it is more specific, captures the purpose to model the correlation within the repeated measures over time, and aligns with regulatory documentation.	–

Reviewer 3 comments	Author response	Changes made
6. As a last minor point, for comment 6, it seems to me that small details like using REML (restricted or residual maximum likelihood) could easily enough be added to the manuscript without overwhelming the reader (e.g., as parenthetical comments).	The text has been amended as suggested, with clarification regarding the use of REML.	Addition to the Methods in bold: “Change from baseline in the EORTC QLQ-C30 on-treatment scores was analyzed using a mixed model repeated measures analysis of all postbaseline scores for each visit. The model was analyzed using restricted maximum likelihood estimation and included treatment arm, visit, and treatment by visit interaction and stratification factors as explanatory variables, and the baseline score and baseline score by visit as covariates.”